# Understanding the Physiological and Molecular Basis for Differences in Nitrogen Use Efficiency in the Parents of a Winter Wheat MAGIC Population

**DOI:** 10.3390/plants13233331

**Published:** 2024-11-28

**Authors:** Aleksander Ligeza, Matthew J. Milner

**Affiliations:** NIAB, 93 Lawrence Weaver Road, Cambridge CB3 0LE, UK

**Keywords:** nitrogen, wheat, MAGIC, grain protein, N uptake

## Abstract

The need to improve both the cost of food production and lower the environmental impact of food production is key to being able to sustainably feed the projected growth of the human population. To attempt to understand how to improve yields under lower nitrogen (N) inputs, a diverse set of UK winter wheats encompassing ~80% of the genetic diversity in current winter wheats in the UK were grown under a range of N levels and their performance measured under various levels of N. This population has parents which encompass all four end-use categories to understand how breeding for differences in NUE may change across different end-use types of wheat. The growth of the eight parents of a MAGIC population showed significant differences in biomass per plant, ear number, yield and protein content of the grain when grown with differing levels of N. No consistent response to N was seen for the lines tested for all of the traits measured. However, the underlying difference in response to N was not due to N uptake or N translocation, as short-term ^15^N uptake and translocation showed no significant differences in the lines tested. RNASeq was then performed on two different bread-making varieties grown under low-N conditions to identify putative genes controlling the underlying differences seen in biomass production when grown on low N. This led to the identification of the genes involved in growth and C/N signaling and metabolism, which may explain the differences in growth and biomass production seen between the parents of this population.

## 1. Introduction

Wheat (*Triticum aestivum* L.) is one of the most consumed plants worldwide, being planted on over 200 million hectares annually and thought to provide both 20% of the global caloric intake as well as 20% of the total protein consumed [1]. To help maintain high agricultural production and limit the unwanted effects of sustainable intensive agriculture production, a greater understanding of how various wheat varieties respond to limited inputs is key.

The largest input for wheat production is typically nitrogen (N), with the cost of the fertilizer accounting for about ~35% of the total cost of production and accounting for ~70% of the greenhouse gas emissions associated with production [2,3]. Furthermore, it has been estimated that only 30–35% of the added N fertilizers are taken up and used by wheat plants in each growing cycle, and the remaining 65–70% are lost to the environment [4]. To start to understand how the underlying genetic differences effect NUE, more knowledge about the physiological and transcriptomic changes that regulate differences in NUE in contrasting varieties is needed. This includes an understanding of how particular varieties with different end uses respond to major inputs such as nitrogen (N) to breed improved varieties.

While assessments of various lines have been tested under field conditions, a general lack of improvement in our understanding of the genes involved in improving NUE still exists [5,6,7,8,9,10,11]. This has led to slow improvements in NUE as it is difficult as to improve NUE because it is a complex trait with many contributing processes including the type of N applied, the amount and timing of fertilization, its interaction between the plant and the soil type, weather, genotype and other management factors affecting its growth. To further complicate the effort to improve NUE, NUE can be further divided into two components: (i) N uptake efficiency (crop N uptake per unit of N available from the soil and fertilizer; NUpE) and (ii) N utilization efficiency (grain DM yield per unit crop of N uptake at harvest; NUtE). Studies have shown that in spring wheats, increased NUE was related mainly to improvements in NUpE, and improved NUE in winter wheats was correlated with increased NUtE [12,13,14,15]. These findings support the idea that variations in NUE can include differences in uptake, perception or utilization of N in different forms, which can help reduce the cost of production and limit the environmental impact of food production. However, these studies did not identify candidate genes underlying the differences in uptake or utilization seen. With the advent of major advancements in sequence information and publicly available mapping populations, this can happen at both the whole-plant and molecular level [16,17].

In the UK, there are four major end classes for which wheat is grown. These classes are called Group 1 through 4. Group 1, which is bread-making wheats, must contain at least 13% protein, have a falling number of 250 s HFN and specific weights of 76 kh/hL, and provide consistent milling and baking performance. The second class is Group 2, which is classed as having bread-making potential but lacks the consistency for milling and baking required for Group 1. Varieties in this class suit specialist flours if certain other baking criteria are met. Group 3 is classed as soft wheats for biscuits, cakes etc. and contains little protein and an extensible but not elastic form of gluten. Finally, Group 4 is listed as feed wheats used for animal feed. As there are several uses for wheat, understanding the physiological and molecular differences which make up various classes is important for breeders when determining which varieties to release and what farmers choose to grow.

Many key genes involved in uptake, translocation and regulation have been identified and characterized in plants. In wheat, the genes believed to be involved in these traits are based only on sequence homology and not characterization of the genes themselves. Thus, most of what is suggested to improve NUE in wheat is based on characterization in other species or changes in expression in other species [18,19,20,21]. This includes four families of transporters involved in the uptake of nitrate, namely NPF/NRT1 (nitrate transporter 1/peptide family), NRT2 (nitrate transporter 2), CLC (chloride channel) and SLAC/SLAH (slow anion channel-associated homologues), as well as ammonium uptake transporters from the AMT family (ammonium transporter). Recently, a total of 292 genes were identified in the wheat genome compared to only 46 NRT2 genes [20]. However, how all of the transporters themselves are regulated and the broad regulation of growth patterns are still poorly understood. But the recent finding that an underlying SNP in NFP2.12 can increase root growth and N absorption under low-N growth conditions shows that using a diverse set of material is key to finding alleles which are important for better NUE [22]. There have also been other genes recently identified including the N transporter TaNRT2.1-6B in wheat which can improve NUE; these include uptake and translocation genes which are important to efficiently acquire the N being applied to the crop [23]. But other genes not involved directly with uptake or N translocation have also been shown to play a role in NUE; this includes a key gene in flowering and development, TaVRN-A1 [11]. A pathway which has been identified is the PII pathway which is highly conserved in many kingdoms from bacteria to plants and believed to regulate the Carbon (C)/N balance in plants. These highly conserved proteins are still a bit of a mystery in terms of how they regulate and incorporate C and N levels in plants. But it is thought that ATP/ADP and 2-oxoglutarate are two of the main metabolites which directly bind the protein to regulate C/N balance [24]. However, depending on the species studied, it is thought that many different metabolites can play a role in PII activity [24]. In Arabidopsis, the PII protein is localized to the chloroplast and their transcripts are regulated by various forms of N and sugar concentrations telling the plant how fast to grow. Overexpression leads to increased anthocyanin production under certain low-N conditions [25,26].

Here, we set out to understand the physiological response of the parents of a MAGIC mapping population, which cover all four end uses of wheat in terms of their ability to grow, on a range of levels of N. The physiological and transcriptome data can then be used to identify candidates at the gene levels for improved NUE. By understanding what traits respond to N in a diverse genetic background, we can start to understand the underlying genes to achieve improved NUE.

## 2. Results

### 2.1. Agronomic Differences in a Diverse Panel of UK Grown Wheats’ Response to Varying Levels of N

To understand how diverse classes of bread wheat respond to varying levels of N, eight varieties which are the parents of a MAGIC mapping population were grown in pots on low-N soil supplemented with four levels of ammonium nitrate to simulate 70, 140, 210 or 280 kg/ha equivalent N levels. Variation was seen in the amount of biomass, yield, tiller number and response to increasing levels of N in the eight varieties tested. Two varieties stood out as generally having significantly better biomass and yield per plant under the range of N levels tested. These two lines Alchemy and Xi19 showed significantly higher levels of above-ground biomass under three of the four N levels tested (Figure 1A). Under the lowest N treatment, most of the varieties showed very little differences in above-ground biomass. Alchemy was significantly higher than Soissons, whereas Xi19 produced significantly higher above-ground biomass compared to all lines except Alchemy and Robigus (*p* val. < 0.05) (Figure 1A). As N levels increased, both Alchemy and Xi19 showed significantly more biomass than all lines, excluding each other, when grown under 140 and 210 kg/ha. At 280 kg/ha equivalents, only Alchemy showed significant increased biomass compared to Brompton and Rialto, with all other varieties showing similar levels of biomass produced (*p* val. < 0.05). Yield per plant mirrored that of biomass production, with significantly increased yields seen in Xi19 relative to Claire, Hereward, Rialto and Soissons at 70 kg/ha equivalents (Figure 1B). Under 140 and 210 kg/ha equivalents, Alchemy and Xi19 showed significantly higher yields than all other lines, excluding each other. Under the highest N level, three lines, Alchemy, Xi19 and Robigus, showed higher yields relative to Brompton.

As seen in Figure 2, with increased nitrogen, some lines increased the number of tillers at different rates than others. Under the lowest N level (70 kg/ha equiv.), no significant differences in the tiller number were observed between any of the eight lines tested. Under 140 kg/ha equivalents, Alchemy and Soissons showed significantly increased tillering relative to Brompton, Claire and Hereward (*p* val. < 0.05). At 210 kg/ha, Alchemy showed increased tillering relative to Brompton, and Soissons showed increased tillering relative to Brompton, Rialto and Robigus. Under the highest N level, Alchemy showed higher tiller numbers than Claire and Rialto. Robigus also showed increased tillering relative to Claire, and Soissons showed significantly higher tillering than all other lines including Alchemy (*p* val. < 0.05). To understand if there were differences in the ability of the varieties to take up N from the soil, the N uptake and translocation at Zadok stage 13 were measured among the eight wheat lines using ^15^N as a tracer. Measurements of uptake and translocation in short-term uptake from the sand did not significantly differ between any of the lines tested when using ^15^N as a tracer (Figure 3A,B).

Further analysis of the lines did however show significant variation in the amount of N in the grain at maturity. Under the lowest N level, seeds from Soissons showed significantly higher grain protein content (GPC) than all other lines tested (Figure 4). The only other line to show significantly higher grain protein content was Rialto compared to Alchemy and Xi19. Under 140 kg/ha N levels, Hereward and Soissons had significantly higher GPC than all other lines, excluding each other. Rialto also showed significantly higher GPC relative to Alchemy and Claire (*p* val. < 0.05). At 210 kg/ha, only Soissons had higher GPC than Alchemy (*p* val. < 0.05). At the highest level of N applied, Claire had significantly lower GPC than Soissons and Brompton. All other comparisons were not significant using a cut-off *p* value of 0.05.

We aimed to understand how the yield, biomass and protein content all influenced the harvest index (HI) and ultimately the NUE (Figure 5). Overall, there were only a few significant differences in the harvest index between the lines themselves at the same N level tested (Figure 5A). Hereward had a significantly lower harvest index than Brompton, Robigus and Xi19 when comparing the lines themselves across all treatments. This was mainly driven by Brompton’s high HI for the top three N levels compared to Hereward. Meanwhile, the differences seen between Hereward and Robigus and Xi19 were driven by the lowest N treatment (70 kg/ha), showing a significantly higher HI than Hereward. Each line did have significantly different changes in the harvest index as N levels increased but this varied by line and treatment.

NUE by comparison showed some interesting trends as expected; the highest NUE was seen under the lowest N treatment, and the general trend was lower NUE as the N level supplied increased (Figure 5B). At the highest N level, the increased grain yields were enough to increase the NUE of Hereward, Rialto, Robigus and Soissons at 280 kg/ha versus that of 210 kg/ha. Meanwhile, the highest yielding lines Alchemy and Xi19 and the lowest yielding line Brompton showed either no significant difference at the two highest levels or significantly lower NUE at the highest N level. There were significant differences in NUE between the lines as well, unlike the HI. Alchemy and Xi19 had significantly different NUE compared to all lines tested, whereas Brompton, Claire, Herward, Rialto, Robigus and Soissons did not have significantly different NUE compared to any lines other than Alchemy and Xi19.

### 2.2. Gene Expression

To identify some of the underlying genes which may be involved in the variation in traits observed in relation to the amount of N supplied, RNASeq was performed to compare the two varieties that contrasted in both biomass and yield but are in the same end use category. The comparison of shoot expression in Hereward and Xi19, both bread-making quality lines, showed a few differences in transcript levels in shoots grown under low N. Overall, 517 genes were differentially expressed between the two varieties, with 216 showing higher expression in Xi19 and 301 showing higher expression in Hereward (Appendix A). The comparison of pathway differences using AgriGO showed significant differences in the biological process, molecular function and cellular component ontologies [27]. The significantly associated terms are shown in Appendix A. Many GO terms are associated with photorespiration, organonitrogen compounds and the chloroplasts amongst the differentially expressed genes. To further reduce the redundancy of the GO terms, REVIGO was used to create a summary of the terms into meaningful pathways (Figure 6). We took the top fifty terms for GO function and this returned 34 updated GO terms showing how differences were observed in energy metabolism, amino acid synthesis and small-molecule synthesis between Hereward and Xi19. One such obsolete term is organonitrogen compound which has been replaced by response to N compound, but this is now the 221st most significant term relative to the 2nd using AgriGO. This result also highlights photorespiration, which might ultimately be the connection between the differences between the biomass produced and the overall differences in the yield observed.

It should be noted the no direct N transporters were identified as differently expressed in the two varieties. A number of amino acid synthesis pathways are differentially regulated, including genes involved in the synthesis of serine, cysteine, lysine and aspartate. However, a PII-type protein, TraesCS1A02G076100.2, annotated as Nitrogen regulatory protein P-II-like protein was significantly more highly expressed, by more than 6-fold, in Hereward compared to in Xi19. Further analysis of homoeologues of TraesCS1A02G076100, TraesCS1B02G094600 and TraesCS1D02G078500 showed that they were not significantly differentially expressed. Other associated genes with C/N homeostasis include two 2-oxoglutarate (2OG) and Fe(II)-dependent oxygenase superfamily proteins on chromosomes 7D and 2A which were also more highly expressed in Hereward relative to Xi19.

## 3. Discussion

The way in which plants respond to nitrogen is not fixed and is related to several underlying genetic differences. While several genes mainly involved in increased uptake or translocation have been shown to increase NUE in wheat, a basic understanding of the differences in the underlying genetic background is key to improving NUE in any given variety [11,22]. Here, we set out to show that the response to N is varied in a mapping population and this includes several key traits associated with yield. From this, we have shown that each end class of wheat studied does not show a consistent phenotypic profile for all of the traits measured, suggesting that other genetic factors outside of N uptake and translocation influence many traits associated with increased yields and improved NUE. This can include grain protein content as Soissons showed significantly higher grain protein content than Xi19, Hereward and Rialto, which are all end-use class 1 or 2, under the two lowest N inputs, even though these four varieties can be used for bread making. In total, five of the eight varieties were able to reach the 13% protein content under the highest N level, showing the plasticity of some of the metrics used for determining end use in wheat. While all eight lines tested showed the typical increase in biomass and yield with increased supplied N (Figure 1), this response varied within each variety tested. Xi19 showed superior biomass production under lower levels of N compared to another six of the eight varieties. This mirrored the average yield per plant, with those having the least biomass showing the lowest yield.

Overall, significant variation was observed in these eight lines for every trait measured apart from N uptake and translocation. This included static traits such as biomass, yield and tiller number. But how each trait varied with different levels of nitrogen was also significantly different between the lines. Taking tillering as an example, Brompton did not add tillers as N increased until the highest N levels but showed increased yields as N levels increased (Figure 2). Soissons, for example, rapidly added extra tillers with each additional supplement of N, but both varieties saw increased yields as N levels increased. Xi19 finally matched Claire in terms of tiller number but out-yielded Claire in the lowest three N treatments. This shows that tillering can correlate well with yields for Xi19, Soissons and Claire, but not for Brompton.

It was interesting that when short-term uptake and short-term translocation were measured, none of the eight lines tested showed any significant differences, suggesting that at least in these varieties, uptake does not limit many of the ways to acquire N. This is in contrast to more long-term measurements obtained by other groups which have shown that five of the eight varieties have differences in final N levels and in the uptake of N from the soil, showing differences in N concentration in the straw and grains when grown under field conditions [5]. Meanwhile, in other species, longer uptake times of 24 h showed significant differences in non-GM material. But for some transgenic plants, an uptake time of only five minutes was enough to see significant differences [28,29,30]. A better understanding of N flux is still needed in a diverse set of lines to understand and tease out varietal differences in how N partitioning effects NUE and GPC. The data presented here suggest that at least for this population, nitrogen utilization is a bigger factor in improving NUE in British wheats than uptake efficiency. This is also supported by previous trials showing that in winter wheats grown in the field, uptake efficiency is not the main cause of differences in NUE. So, further studies in spring wheat might help support if uptake from the soil can be improved [13,14].

More surprising was the levels of protein in the grain of the different lines. Alchemy was relatively insensitive to external N supply and did not achieve protein levels above 9% until the highest N treatment. Brompton, Claire and Hereward were insensitive to N in the middle of the range, closest to what a farmer typically adds to a field in the UK. Rialto and Robigus were insensitive at the low end of the treatments and showed slightly increased protein levels at 210 kg/ha and increased GPC by more than 2% under the highest levels. Finally, Soissons and Xi19 showed increased GPC incrementally with each added N application. These data suggest that there is variation in the way in which a plant perceives its N status and how much N can be remobilized to the grain. We still do not understand how wheat or any plant for that matter perceives N or the pathways that effect the decision by the plant to remobilize the N to the grain

To understand why similar end-use varieties behaved so differently, a comparative analysis of the differentially expressed genes was performed for bread-making quality varieties. To further support the other previous data collected, no N regulated transporters or known components such as glutamine synthetase/glutamate synthase (GS) were identified as differentially expressed between the two varieties. However, the identification of a PII-type protein and 2-oxoglutarate showing higher expression in Hereward suggests that some of the differences in biomass are the result of photosynthesis and C signaling rather than due to transporting enough N to the leaves to maintain photosynthesis. These same lines have been shown to have differing levels of NH_4_ in their tissues and differing responses to N in the short term, including differences in GS activity [31]. These data do have to be taken with some skepticism as short-term gene changes may not be the most predictive for longer term differences in N levels observed. In many bacteria, GS activity is post-transcriptionally regulated by the PII protein, one of the most widely distributed signal transduction proteins coordinating N and C levels to maintain growth [24]. In turn, PII is allosterically regulated by ATP/ADP and 2-OG, while GlnD, which has a glutamine-binding motif, regulates PII activity in response to cellular glutamine concentration by uridylation [24]. These are most likely key sinks of both C and N which can be easily monitored for feedback inhibition rather than the elements themselves. Thus, bacterial C and N metabolisms are controlled by GS/GOGAT through the integration of information from a signaling network—consisting of sensory, signaling and regulatory proteins under allosteric or post-transcriptional control—that can rapidly respond to internal and environmental changes. In contrast, quite a bit less is known about plants, but in Arabidopsis, a PII-like protein has a role in regulating the ornithine/arginine synthesis pathway in a glutamine-dependent manner [32]. While again we did not see differentially expressed genes involved in glutamine synthesis, we did see several amino acid pathways differentially regulated, maybe suggesting that other amino acids can also regulate PII-type proteins. Either way, large differences in C/N signaling were observed between Hereward and Xi19, which could be the underlying reason for the differences seen in biomass and yield between the lines. However, the proteins themselves, while highly conserved in many different branches of life, are dispensable as a loss of function in Arabidopsis showed no obvious growth impairment relative to WT plants [33]. The insights gained from understanding the biochemical mechanisms used by PII proteins to modulate central metabolism can hopefully help to enhance both yield and resilience to abiotic stresses.

In conclusion, NUE improvements will most likely involve a better understanding of genes other than just nitrogen transporters, and other physiological improvements in yields and plant performance can also improve NUE to help reduce the cost and environmental conditions of global food production.

## 4. Materials and Methods

Plant growth conditions: Each line was grown on TS5 low-fertility soil to control total nitrogen, with a starting nitrogen level of 0.1 mg/L (Bourne Amenity, Kent, UK). The soil was roughly 17% clay, 58% sand and 25% silt with a pH of 7.5. The soil was supplemented with super phosphate at a rate of 30 mg per pot. Ammonium nitrate was then added to each pot to reach a final concentration in the pots equivalent to field fertilizer application of 70, 140 or 210 kg N/ha. Plants were grown in a climate-controlled glasshouse with 10,000 lx sodium supplemental light for a 16 h day and 20 °C/15 °C day/night temperatures. At least 21 plants per line were grown per treatment in a randomized pot design. Plants were watered daily for the entire lifecycle.

RNA extraction and analysis: For transcript abundance measurements and sugar content measurement, wheat seedlings were grown for ten days in 2.2 L pots containing Magnavaca solution composed of 3 µM KH_2_PO_4_, 3.52 mM Ca(NO_3_)_2_, 0.58 mM, KCl, 0.58 mM K_2_SO_4_, 0.56 mM KNO_3_, 0.86 mM Mg(NO_3_)_2_ 0.13 mM H_3_BO_3_, 5 µM MnCl_2_, 0.4 µM Na_2_MoO_4_, 10 µM ZnSO_4_, 0.3 µM CuSO_4_ Fe(NO_3_)_3_ and 2 mM MES (pH 5.5), supplemented with 0.4 mM NH_4_NO_3_. RNA was extracted using an RNeasy Kit (Qiagen). RNASeq libraries were generated for using Illumina HiSeq2500 paired-end reads. The cDNA libraries were treated with the enzyme Ribo—Zero (Illumina) to reduce the abundance of ribosomal RNAs before the libraries were run on Illumina HiSeq2500. Sequencing was performed by the Earlham Institute. Reads of the RNA samples were analyzed by mapping the transcript levels using Salmon v 1.10 to the cDNAs of wheat RefSeq v 1.1 [34]. The reads were quantified using edgeR after removal of cDNAs which had less than 10 total mapped reads across the six libraries [35]. A maximum FDR of 0.05 was used as a test of significance. The reads were deposited in the ENA under project number PRJEB79052. GO term enrichment was based on AgriGO v2.0, using differentially expressed genes between the two varieties [27]. Redundant and outdated terms were filtered using REVIGO to improve readability [36].

N level determination in wheat grains: Samples were measured using the Dumas method for N determination. The samples were dried for 17 h at 100 °C and then milled on a 1 mm hammer mill. Prior to testing, the samples were dried at 104 °C for 3 h and 1 g of sample was loaded on the instrument (Leco TruMacN Dumas gas analyzer), following the manufacturer’s instructions. The samples were converted to gases by heating in a combustion tube at 1150 °C. Interfering components were removed from the resulting gas mixture. The nitrogen compounds in the gas mixture or a representative part of the mixture were converted to molecular nitrogen which was quantitatively determined by using a thermal conductivity detector. The nitrogen content was then calculated by using a microprocessor. To calculate the percentage of protein, the standard ISO conversion factor of 5.7 was applied.

^15^N measurements: To measure N uptake, roots from ~2-week-old seedlings at Zadok stage 13 were exposed to ^15^NH_4_^15^NO_3_ for 5 min, then washed in 0.1 mM CaSO_4_ for 1 min, harvested and dried at 70 °C for 48 h. The samples were analyzed for percentage nitrogen, ^14^N/^15^N (δ^15^N), using a Costech Elemental Analyzer attached to a Thermo DELTA V mass spectrometer in continuous flow mode. The excess ^15^N was calculated based on measurements of δ^15^N and tissue N%. First, the absolute isotope ratio (R) was calculated for labeled samples and controls using R_standard_ (the absolute value of the natural abundance of ^15^N in atmospheric N2).
R_sample or control_ = [(δ^15^N = 1000) + 1] × R_standard_

Then, the molar fractional abundance (F) and mass-based fractional abundance (MF) were calculated:F = R_sample or control_ = (R_sample or control_ + 1)
MF = (F × 15)/[(F × 15) + ((1 − F) × 14)]
ΔMF = MF_sample_ − MF_control_

The excess ^15^N in mg in the total tissue was calculated as follows:Excess ^15^N(g) = ΔMF × Tissue dw(g) × Tissue N%/100

## Figures and Tables

**Figure 1 plants-13-03331-f001:**
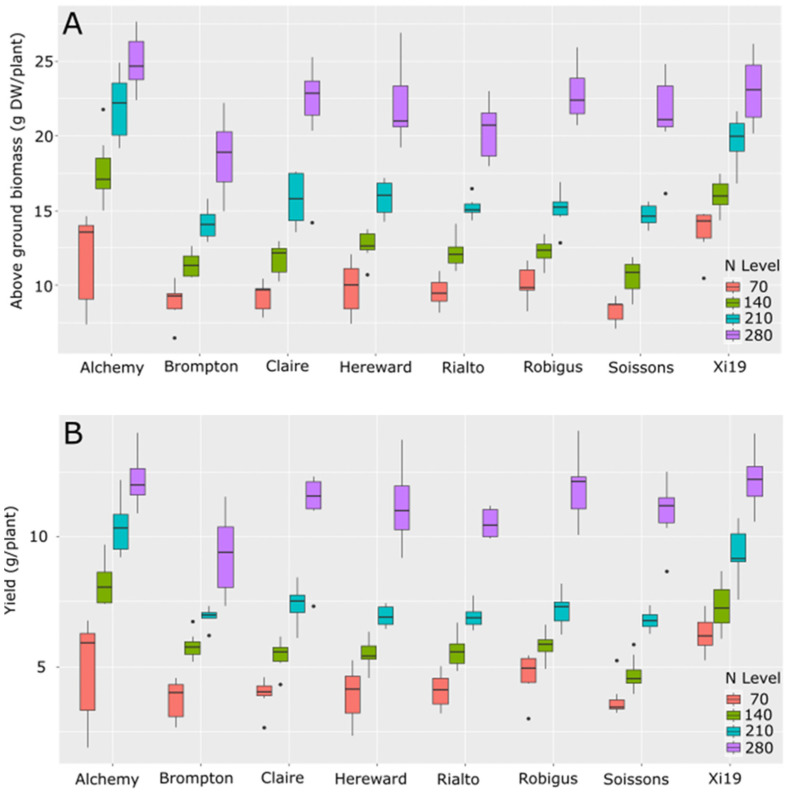
Performance of growth for founders of MAGIC population under four different levels of N measured for above-ground biomass (**A**) and yield per plant (**B**). Dots represent values outside of the Quatile±1.5*IQR.

**Figure 2 plants-13-03331-f002:**
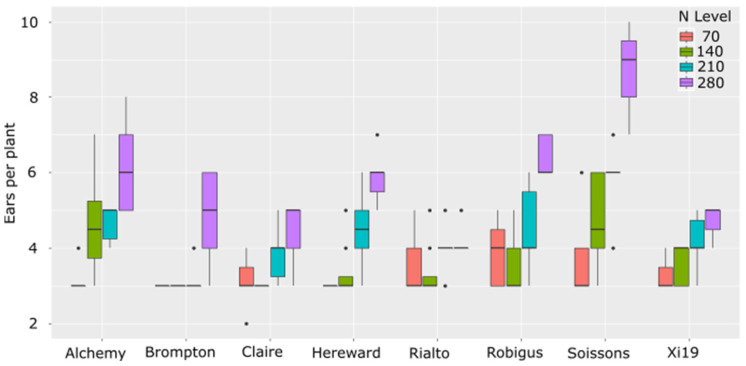
Ear per plant in founders of MAGIC population grown under four levels of N. Dots represent values outside of the Quatile±1.5*IQR.

**Figure 3 plants-13-03331-f003:**
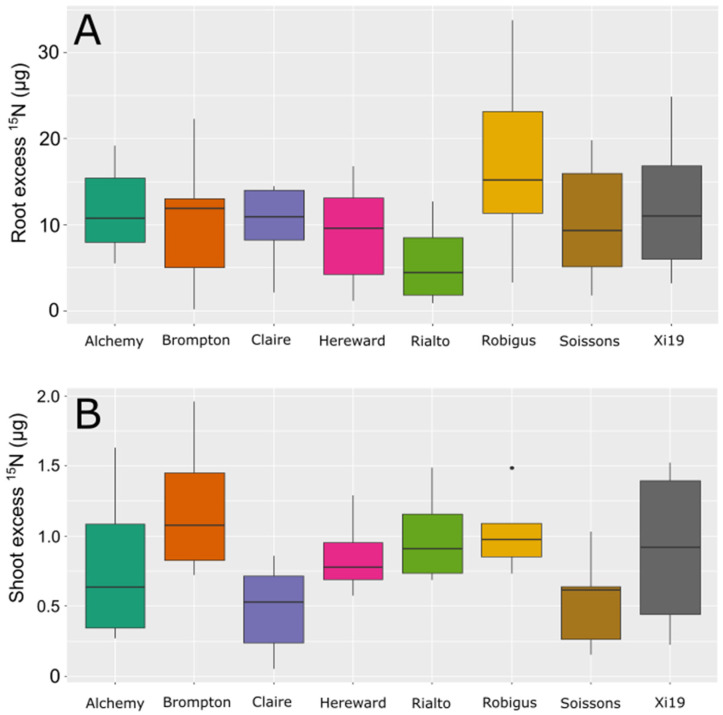
Nitrogen uptake (**A**) and translocation (**B**) in founders of MAGIC population using ^15^N tracer. Dots represent values outside of the Quatile±1.5*IQR.

**Figure 4 plants-13-03331-f004:**
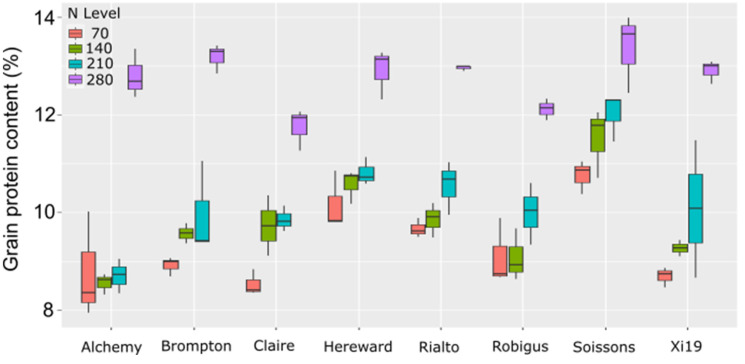
GPC in founders of MAGIC population grown under four different N levels.

**Figure 5 plants-13-03331-f005:**
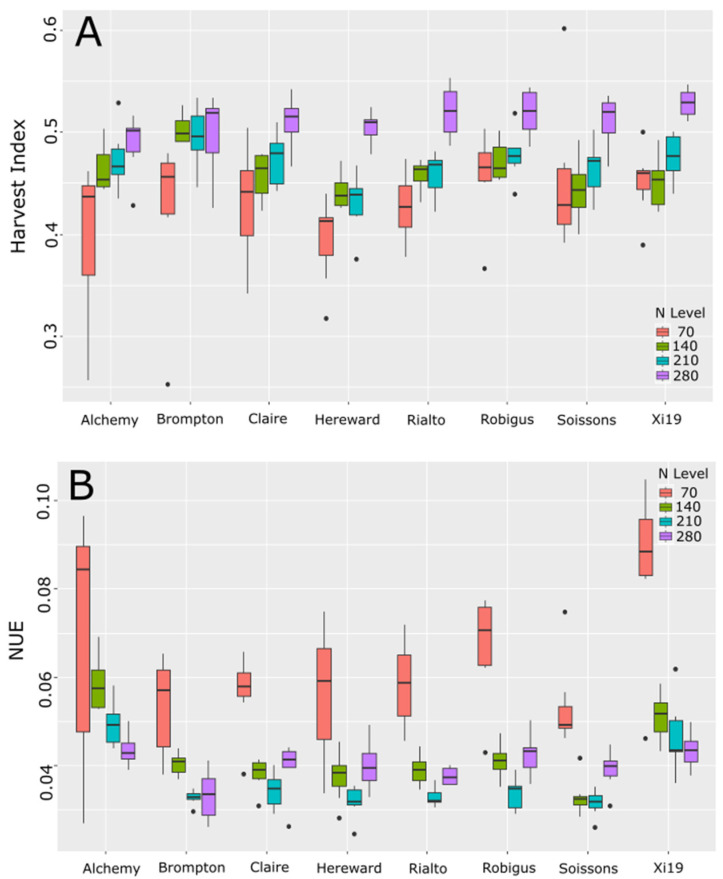
Harvest index (HI) (**A**) and nitrogen use efficiency (NUE) (**B**) of eight varieties tested under four different levels of N. Dots represent values outside of the Quatile±1.5*IQR.

**Figure 6 plants-13-03331-f006:**
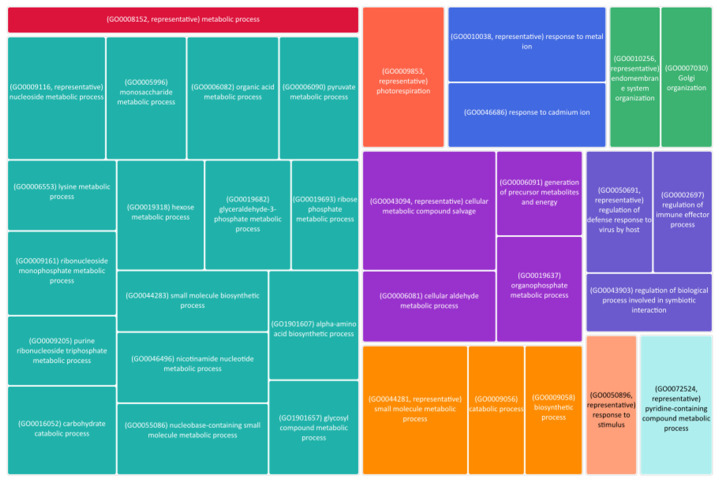
Revised GO terms associated with the differentially expressed genes between Herward and Xi19.

## Data Availability

The RNAseq raw reads were deposited in SRA under project code PRJNA1163677.

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
