# Peer review of "Understanding the Physiological and Molecular Basis for Differences in Nitrogen Use Efficiency in the Parents of a Winter Wheat MAGIC Population"

_plants, 2024, doi:10.3390/plants13233331_

Round 1
Reviewer 1 Report
Comments and Suggestions for Authors
Nitrogen does play a huge role in ensuring global food security. However excessive use of nitrogen fertilizer may also bring potential environmental risks. Current studies have shown that excessive use of nitrogen fertilizer can inhibit plant growth instead of increasing the absorption of nitrogen fertilizer by plants. Therefore it is particularly important to use nitrogen fertilizer reasonably. However the data on the utilization efficiency of different wheat varieties to different levels of nitrogen fertilizer are still lacking. The study used different cultivars to test the performance of agronomic traits at different levels of nitrogen fertilizer utilization, I think this research will attract a lot of interest. I hope this article can further improve in the following aspects:
1. The use of different colors in Figure 3 makes it easy to confuse with the colors used in the previous figures, It is recommended to use the same color, In addition A and B are best placed in the upper left corner of the graph.
2. Different color descriptions should be added to Figure 4.
3. Results of Part 2, Analysis of Gene Expression Differences among Varieties, Tables S1 and S2 in the supplenentary should be included in the text in the form of graphics or diagrams rather than in supplenentary. The data in this section remains an important part of the manuscript of this paper. Should be shown and presented in more detail.
4. It is recommended to cite relevant literature in recent years.
Author Response
- The use of different colors in Figure 3 makes it easy to confuse with the colors used in the previous figures, It is recommended to use the same color, In addition A and B are best placed in the upper left corner of the graph.
We have changed the colors of Figure 3 as requested to no confuse the reader about what is being measured in terms of treatments versus lines tested in the other figures. A and B have also been moved to the upper left as requested by the reviewer.
- Different color descriptions should be added to Figure 4.
Sorry for the omission. A figure legend has been added.
- Results of Part 2, Analysis of Gene Expression Differences among Varieties, Tables S1 and S2 in the supplementary should be included in the text in the form of graphics or diagrams rather than in supplementary. The data in this section remains an important part of the manuscript of this paper. Should be shown and presented in more detail.
We have added a figure with the top fifty GO terms to highlight the differences seen between the two lines tested and show the pathways which differ. We feel that the data is important but a list of 500 genes is not a meaningful table in the text. We hope this helps highlight the differences observed between the two varieties.
- It is recommended to cite relevant literature in recent years.
We have added to the introduction and discussion to further show how the work fits into the more recent work on the subject matter. We hope the reviewer finds this acceptable or we can highlight a certain aspect of the literature if this does not suffice.
Reviewer 2 Report
Comments and Suggestions for Authors
This manuscript investigates the differential response of different parents to nitrogen fertilizer in the MAGIC population of winter wheat in the UK, and analyzes the underlying genetic mechanisms through physiological and transcriptome data. This study has certain innovation and application value, providing a theoretical basis for genetic improvement of nitrogen fertilizer utilization efficiency in wheat.
For this manuscript, specific suggestions are as follows,
1. Suggest further refining the summary, for example, there are significant differences in the response of different parents to nitrogen fertilizer, what are the specific results and data. Highlight the potential gene PII protein and 2-ketoglutarate-dependent ferritin superfamily protein related supporting data discovered through research.
2. Results and Discussion. Gene expression analysis: It is recommended to further explain the mechanisms of PII protein and 2-ketoglutarate-dependent ferritin superfamily proteins in wheat nitrogen fertilizer utilization efficiency, and explore their interactions with other nitrogen related genes.
3. Suggest combining physiological data and gene expression data to deeply analyze the impact of carbon and nitrogen signaling pathways on nitrogen fertilizer utilization efficiency in wheat.
4. Supplement the research conclusion of this manuscript.
Comments on the Quality of English LanguageThe language and logical structure of this article are good.
Author Response
- Suggest further refining the summary, for example, there are significant differences in the response of different parents to nitrogen fertilizer, what are the specific results and data. Highlight the potential gene PII protein and 2-ketoglutarate-dependent ferritin superfamily protein related supporting data discovered through research.
We have further added to the results and discussion to highlight the differences between Hereward and Xi19 to explain the differences in biomass production and further develop the PII proteins in plants and what is currently known. We hope the reviewer finds this version acceptable.
- Results and Discussion. Gene expression analysis: It is recommended to further explain the mechanisms of PII protein and 2-ketoglutarate-dependent ferritin superfamily proteins in wheat nitrogen fertilizer utilization efficiency, and explore their interactions with other nitrogen related genes.
We have added to the discussion about the genes in wheat and possible mechanisms of how PII type proteins might cause differences in plant biomass production and ultimately NUE.
- Suggest combining physiological data and gene expression data to deeply analyze the impact of carbon and nitrogen signaling pathways on nitrogen fertilizer utilization efficiency in wheat.
We agree this is a good idea but not in the scope of the current work. Carbon and nitrogen isotopes can be used to understand flux data and N movement but again this is substantial additional work and we feel this is outside of this particular work.
- Supplement the research conclusion of this manuscript.
We have added to the discussion to highlight the physiological responses and how certain genes and pathways might influence NUE in the lines tested. We hope the reviewer finds this acceptable.
Reviewer 3 Report
Comments and Suggestions for Authors
The manuscript deals with the response to different nitrogen levels of a set of 8 wheat genotypes, used as parental of a Multiparent Advanced Generation Inter-Cross (MAGIC) population. The manuscript, per se, is well written and well discussed. However, it is clear that this is a preliminary characterization for the further evaluation of nitrogen use efficiency of the MAGIC population obtained. The manuscript is quite essential and presents different flaws. The graphs report the box-plots of the main productive traits, including grain protein content, grouped with the different genotypes at the different nitrogen rates. A large transcriptomic dataset in reported in supplementary. The actual version of the manuscript does not apport novelty to the literature as presented. If the authors decide to report the results of the current investigation without any indication of the performance of the MAGIC population, the major suggestion is to improve the current version with more statistical analysis. First of all, no NUE trait was reported, and it should be mandatory, since the aim proposed in the title. Differences in NUE components and nitrogen harvest index could contribute to interpreting the differences, providing indications for the future. This should be also better analyzed, not only with descriptive statistics, but also to highlight the contribution of a specific genotype in the population. In addition, some multivariate analysis would also contribute to linking the transcriptomic observations with the agronomic traits investigated. I think that the current dataset deserves a better effort, since the topic is relevant for scientists (genetics, physiology, agronomy etc…) and for breeders. Alternatively, the authors could consider directly publishing their dataset in dedicated service journal. For these reasons, this manuscript is suggested to be resubmitted after a strong revision setting, according to the indications provided. This is for the valorize the great job conducted by the authors.
Minor suggestions: better details of the growing conditions of the trials are necessary (soil, water, temperature etc.…). In addition, yield provided per plant does not take into account the tillering potential and can lead to flaws in order to understand NUE response, since N dilution could occur because of plants and per grains density
Author Response
The manuscript deals with the response to different nitrogen levels of a set of 8 wheat genotypes, used as parental of a Multiparent Advanced Generation Inter-Cross (MAGIC) population. The manuscript, per se, is well written and well discussed. However, it is clear that this is a preliminary characterization for the further evaluation of nitrogen use efficiency of the MAGIC population obtained. The manuscript is quite essential and presents different flaws. The graphs report the box-plots of the main productive traits, including grain protein content, grouped with the different genotypes at the different nitrogen rates. A large transcriptomic dataset in reported in supplementary. The actual version of the manuscript does not apport novelty to the literature as presented. If the authors decide to report the results of the current investigation without any indication of the performance of the MAGIC population, the major suggestion is to improve the current version with more statistical analysis. First of all, no NUE trait was reported, and it should be mandatory, since the aim proposed in the title. Differences in NUE components and nitrogen harvest index could contribute to interpreting the differences, providing indications for the future. This should be also better analyzed, not only with descriptive statistics, but also to highlight the contribution of a specific genotype in the population. In addition, some multivariate analysis would also contribute to linking the transcriptomic observations with the agronomic traits investigated. I think that the current dataset deserves a better effort, since the topic is relevant for scientists (genetics, physiology, agronomy etc…) and for breeders. Alternatively, the authors could consider directly publishing their dataset in dedicated service journal. For these reasons, this manuscript is suggested to be resubmitted after a strong revision setting, according to the indications provided. This is for the valorize the great job conducted by the authors.
Response 1: We are sorry the reviewer does not feel the paper is novel. The reviewer is correct the work is setting the stage for further studies to define the genes involved in many of the traits discussed in the presented manuscript. We disagree with the fact that the finds presented are not novel. We show across different end use classes that NUE can vary not by use but by underlying genetics. We also show that between the same end uses some of the genes and pathways which differ and may be involved in traits breeders find valuable to increase production or as the reviewer points out remove unwanted traits such as extra tillers without increased yields. To our knowledge this is not the case. We would also like to point out these lines are being fully sequenced and some of that sequence is not publicly available along with the population itself as being available for any researcher that would like the seeds.
We have as the reviewer suggested added a figure with both the HI and NUE to further support the understanding of how the different lines reacted to the change in nitrogen to show further differences amongst the varieties tested. We how the reviewer finds this acceptable.
Minor suggestions: better details of the growing conditions of the trials are necessary (soil, water, temperature etc.…). In addition, yield provided per plant does not take into account the tillering potential and can lead to flaws in order to understand NUE response, since N dilution could occur because of plants and per grains density.
We have added to the methods of how the plants were grown to further ensure reproducibility of the experimental design.